# Vibration Analysis of a 1-DOF System Coupled with a Nonlinear Energy Sink with a Fractional Order Inerter

**DOI:** 10.3390/s22176408

**Published:** 2022-08-25

**Authors:** Yandong Chen, Yongpeng Tai, Jun Xu, Xiaomei Xu, Ning Chen

**Affiliations:** 1College of Mechanical and Electronic Engineering, Nanjing Forestry University, Nanjing 210037, China; 2College of Intelligent Equipment Engineering, Wuxi Taihu University, Wuxi 214151, China; 3College of Automobile and Traffic Engineering, Nanjing Forestry University, Nanjing 210037, China; 4Anhui Vocational and Technical College, College of Mechanical Engineering, Hefei 230009, China

**Keywords:** fractional order inerter, nonlinear energy sink, equilibrium point, stability analysis, modulated response

## Abstract

The fluid inerter described by the fractional derivative model is introduced into the traditional nonlinear energy sink (NES), which is called fractional-order NES in this paper. The slowly varying dynamic equation (SVDE) of the system coupled with fractional-order NES is obtained by the complex averaging method, in which the fractional derivative term is treated using the fractional Leibniz theorem. Then, the discriminants (Δ, Δ_1_, and Δ_2_) of the number of equilibrium points are derived. By using the variable substitution method, the characteristic equation for judging the stability is established. The results show: (1) the approximate SVDE is sufficient to reflect the slowly varying characteristics of the system, which shows that the mathematical treatment of the fractional derivative term is reliable; (2) the discriminant conditions (Δ_1_, Δ_2_) can accurately reflect the number of equilibrium points, and the corresponding range of nonlinear parameter *κ* can be calculated when the system has three equilibrium points. The expressions of Δ_1_, Δ_2_ are simpler than Δ, which is suitable for analysis and design parameters; (3) the stability discrimination methods of schemes 1 and 2 are accurate. Compared with scheme 2, scheme 1 is more prone to various responses, especially various strongly and weakly modulated responses. In scheme 2, the inertia effect of mass can be completely replaced by integer order inerter. Compared with integer order inerter, the introduction of fractional order inerter, whether in series or in parallel, means that the amplitude of the equilibrium point on the NES vibrator is smaller, but it is also for this reason that it is not easy to produce a modulated response with scheme 2, and the vibration suppression effect of the main structure is not good.

## 1. Introduction

When a fluid inerter works [1,2,3,4,5], it produces inertia force, usually accompanied by large parasitic damping force, which cannot be ignored. Therefore, the traditional mathematical model of an ideal inerter, such as a mechanical inerter [6,7,8], cannot describe the characteristics of a fluid inerter. The approximate model of its output force is mainly obtained by referring to the empirical formula. For example, the output force model of the fluid inerter proposed in [9,10,11] is obtained using the laminar flow empirical formula, and the output force model is established using the turbulence empirical formula in [4,5]. Some of these models consider laminar flow, and others consider the form of turbulence, but the actual situation may be both. There is still room for discussion on the model of fluid inerters. In reference [12], the author noted that describing the multiphase mechanical properties by simply adding the integer derivatives of different derivative orders is not satisfactory. From the perspective of a modeling fractional differential equation, Westerlund [13] proposed the use of a unified fractional model containing multiphase mechanical properties. In reference [14], a fractional derivative model is proposed to describe the output force model of a fluid inerter. It is verified both mathematically and physically that the model can reflect the multiphase mechanical characteristics of the inerter. Compared with the integer order, an inerter, although it brings difficulties in mathematical processing [15], can have more parameter combinations, to ensure that the system achieves the required performance.

In recent years, NES [16,17] became a research hotspot of nonlinear dynamic vibration absorbers. The typical NES is composed of additional mass, a nonlinear spring, and linear damping, which is far superior to the traditional linear vibration absorber. The effective vibration suppression of NES is mainly through the TET [18,19,20], and its internal mechanism is resonance capture. Only when the external excitation intensity reaches the trigger threshold does the TET become excited. Under harmonic excitation, the study of NES systems is accompanied by the discussion of the mechanism of the strongly modulated response (SMR) generation [16,21,22,23,24]. These research results show that the vibration system has better energy transfer efficiency and vibration suppression effect when SMR appears. The SMR is due to the coupling system slowly varying power flow caused by the saddle node bifurcation of limit cycles, and is a real phenomenon in engineering. The saddle node bifurcation is directly related to the stability of the equilibrium point of the slowly varying power flow. Scholars [24,25,26] used a complex average method, multi-scale method, harmonic balance method, and so on, to study the stability of the equilibrium points. References [27,28,29,30] introduce a new idea that parallel or series inertial ideal integer order inerter based on traditional NES can reduce the added mass and improve the vibration suppression performance. However, the fractional order inerter has not been applied to nonlinear vibration absorbers yet. In this paper, we will introduce the fractional order inerter to NES structure and discuss the influence of nonlinear parameters and inerter parameters on the vibration system.

The remainder of this paper is organized as follows. In Section 2, the dynamic models of the two schemes are established. In Section 3, the SVDE, the discriminant of the number of equilibrium points, and the stability discrimination of the equilibrium points are derived for the two schemes. Section 4 is the numerical verification of the theoretical analysis in Section 3. Section 5 contains the conclusions.

## 2. Dynamic Model of a Linear SDOF System Coupled with Fractional-Order NES

The two models of the SDOF system coupled with fractional-order NES are shown in Figure 1, where scheme 1 (S.1) is the NES with an inerter in parallel (a); and scheme 2 (S.2) is the NES with one inerter in series (b). In Figure 1, *m*_1_, *k_s_*, and *c_s_* are the main structural mass, stiffness, and damping coefficient, respectively; *m*_2_, *k_n_*, *c_n_*, and *b_n_* are the mass, ideal nonlinear cubic stiffness, damping coefficient of the NES, and the inertance of the inerter, respectively. Since this paper discusses the influence of the fractional derivative of the fractional order inerter on the research, the influence of nonlinear factors and Coulomb friction is ignored [10,11,14]. The fractional order inerter model is
(1)$$F_{b}=b_{n}D^{\mu }z\;$$
where *μ* between 1 and 2, especially when *μ* = 2, is an ideal integer order inerter. In recent decades, the most used definitions of fractional calculus are the Riemann–Liouville, Caputo, and Grunwald–Letnikov derivatives [31]. Among them, the Caputo definition is suitable for the application of various engineering scenarios. Therefore, Caputo’s definition is adopted for the fractional derivatives in this paper.

Therefore, the dynamic models of the two schemes can be obtained from Newton’s second law, as shown in Equations (2) and (3).

S.1:(2)$$\left\{ \begin{array}{l} m_{s}{\ddot{z}}_{1}+m_{2}{\ddot{z}}_{2}+c_{s}{\dot{z}}_{1}+k_{s}z_{1}=f\cos\omega t\\ m_{2}{\ddot{z}}_{2}-\left[b_{n}D^{\mu }\left(z_{1}-z_{2}\right)+c_{n}\left({\dot{z}}_{1}-{\dot{z}}_{2}\right)+k_{n}{\left(z_{1}-z_{2}\right)}^{3}\right]=0 \end{array}\right.,$$
and

S.2:(3)$$\left\{ \begin{array}{l} m_{s}{\ddot{z}}_{1}+m_{2}{\ddot{z}}_{2}+b_{n}D^{\mu }z_{2}+c_{s}{\dot{z}}_{1}+k_{s}z_{1}=f\cos\omega t\\ m_{2}{\ddot{z}}_{2}+b_{n}D^{\mu }z_{2}-c_{n}\left({\dot{z}}_{1}-{\dot{z}}_{2}\right)-k_{n}{\left(z_{1}-z_{2}\right)}^{3}=0 \end{array}\right.$$

Obviously, when *b_n_* = 0, it is the vibration dynamic model coupled with the traditional NES.

According to reference [32], when the generalized harmonic analysis is performed, the fractional derivative terms in vibration Equations (2) and (3) can be approximately equivalent in the following integer order as follows:(4)$$\left\{ \begin{array}{l} D^{\mu }z_{2}=\omega^{\mu -1}\cos\left(\left(\mu -1\right)\pi /2\right){\dot{z}}_{2}+\omega^{\mu -2}\sin\left(\left(\mu -1\right)\pi /2\right){\ddot{z}}_{2}\\ D^{\mu }\left(z_{1}-z_{2}\right)=\omega^{\mu -1}\cos\left(\left(\mu -1\right)\pi /2\right)\left({\dot{z}}_{1}-{\dot{z}}_{2}\right)+\omega^{\mu -2}\sin\left(\left(\mu -1\right)\pi /2\right)\left({\ddot{z}}_{1}-{\ddot{z}}_{2}\right) \end{array}\right.$$

Here, the first term on the right of the above formula plays a damping role, and the second term plays an inertia role. After substituting (4) into Equations (2) and (3), *m*_2_ cannot be 0 for S.1, but *m*_2_ can be equal to 0 for S.2, and its inertia is completely provided by the inerter.

By introducing a new time scale $\tau =\omega_{0}t,\omega_{0}=\sqrt{k_{s}/m_{s}},\Omega =\omega /\omega_{0}$, the above equation is dimensionless, and the following equation is obtained:

S.1:(5)$$\left\{ \begin{array}{l} {z^{\prime \prime }}_{1}+\varepsilon {z^{\prime \prime }}_{2}+\xi_{s}{z^{\prime }}_{1}+z_{1}=f_{e}\cos\Omega \tau \\ \varepsilon {z^{\prime \prime }}_{2}-\left[\varepsilon_{1}\omega_{0}^{\mu -2}\cdot D^{\mu }\left(z_{1}-z_{2}\right)+\xi_{n}\left({z^{\prime }}_{1}-{z^{\prime }}_{2}\right)+\kappa {\left(z_{1}-z_{2}\right)}^{3}\right]=0 \end{array}\right.,$$
and

S.2:(6)$$\left\{ \begin{array}{l} {z^{\prime \prime }}_{1}+\varepsilon {z^{\prime \prime }}_{2}+\varepsilon_{1}\omega_{0}^{\mu -2}\cdot D^{\mu }z_{2}+\xi_{s}{z^{\prime }}_{1}+z_{1}=f_{e}\cos\Omega \tau \\ \varepsilon {z^{\prime \prime }}_{2}+\varepsilon_{1}\omega_{0}^{\mu -2}\cdot D^{\mu }z_{2}-\xi_{n}\left({z^{\prime }}_{1}-{z^{\prime }}_{2}\right)-\kappa {\left(z_{1}-z_{2}\right)}^{3}=0 \end{array}\right.$$
where $\Omega =\frac{\omega }{\omega_{0}},\;\varepsilon =\frac{m_{2}}{m_{s}},\;\varepsilon_{1}=\frac{b_{n}}{m_{s}},\;\xi_{s}=\frac{c_{s}}{\sqrt{m_{s}k_{s}}},\;\xi_{n}=\frac{c_{n}}{\sqrt{m_{s}k_{s}}},\;\kappa =\frac{k_{n}}{k_{s}},\;f_{e}=\frac{f}{k_{s}}$.

## 3. Equilibrium Point Analysis

### 3.1. The SVDE of the System Coupled with Fractional-Order NES (S.1)

In this paper, the nonlinear term is not limited to a small quantity, so system (5) can be a strongly nonlinear system. The complex averaging method (CA-X) [25,26] is used to derive the SVDE of system (5). Internal resonance of the system is one of the preconditions to ensure TET. Therefore, this paper studies the characteristics of the equilibrium point at 1:1 internal resonance i.e., Ω =1, and introduces complex variables as follows:(7)$$\varphi_{j}e^{i\tau }={z^{\prime }}_{j}+iz_{j},\varphi_{j}^{*}e^{-i\tau }={z^{\prime }}_{j}-iz_{j}(j=1,2).$$

The following relations can be obtained by further derivation of the (7):(8)$$z_{j}=\frac{\varphi_{j}e^{i\tau }-\varphi_{j}^{*}e^{-i\tau }}{2i},{z^{\prime }}_{j}=\frac{\varphi_{j}e^{i\tau }+\varphi_{j}^{*}e^{-i\tau }}{2},{z^{\prime \prime }}_{j}={\dot{\varphi }}_{1}e^{i\tau }+i\frac{\varphi_{j}e^{i\tau }-\varphi_{j}^{*}e^{-i\tau }}{2}.$$

Equation (8) can be further transformed to obtain the following relationship:(9)$${\dot{\varphi }}_{j}e^{i\tau }-{\dot{\varphi }}_{j}^{*}e^{-i\tau }=0,{\dot{\varphi }}_{j}e^{i\tau }+{\dot{\varphi }}_{j}^{*}e^{-i\tau }=2{\dot{\varphi }}_{1}e^{i\tau }$$

According to the properties of fractional calculus [31], $D^{\mu }z_{j}=D^{\mu -1}\left[\frac{dz_{j}}{dt}\right]$, that is, $D^{\mu }z_{j}=D^{\alpha }\left[{\dot{z}}_{j}\right]$, *α* = *μ* − 1, *α* ∈ (0, 1). The fractional Leibniz theorem [31] is introduced as follows:(10)Dtαt0[f(t)g(t)]=∑k=0n(αk)f(k)(t)g(α−k)(t)−Rnα(t), 
where $\left(\begin{array}{c} \alpha \\ k \end{array}\right)=\frac{\Gamma \left(\alpha +1\right)}{k!\Gamma \left(\alpha -k+1\right)}$, Γ(⋅) is the gamma function, and the functions *f*(*t*) and *g*(*t*) and their derivatives are continuous on the interval [*t*_0_, *t*]. Since 0 < *α* < 1, it can be seen from [31] that when *n* = 2 and *k* < 2, the remainder $R_{n}^{\alpha }\left(t\right)$ can be ignored.

Therefore, according to Equations (8)–(10), $D^{\mu }z_{j}$ can be reduced to:(11)$$D^{\mu }z_{j}=D^{\alpha }\left(\frac{\varphi_{j}e^{i\tau }+\varphi_{j}^{*}e^{-i\tau }}{2}\right)=\frac{\left(\varphi_{j}e^{i\tau }+\varphi_{j}^{*}e^{-i\tau }\right)\cos\frac{\alpha \pi }{2}+\left(\varphi_{j}e^{i\tau }-\varphi_{j}^{*}e^{-i\tau }\right)\sin\frac{\alpha \pi }{2}i}{2}+\alpha {\dot{\varphi }}_{j}e^{i\tau }\sin\frac{\alpha \pi }{2}.$$

Then, the Equations (7), (8) and (11) are substituted into Equation (5), and averaged to obtain the SVDE of S.1:(12)$$\left\{ \begin{array}{l} {\varphi^{\prime }}_{1}+\varepsilon {\varphi^{\prime }}_{2}+\xi_{s}\frac{\varphi_{1}}{2}+i\frac{\varepsilon \varphi_{2}}{2}=\frac{f_{e}}{2}\\ \left(\varepsilon +\alpha b\right){\varphi^{\prime }}_{2}-\alpha b{\varphi^{\prime }}_{1}+i\frac{\varepsilon \varphi_{2}}{2}-\left(c+bi+\xi_{n}-\frac{3\kappa }{4}i{\left|\varphi_{1}-\varphi_{2}\right|}^{2}\right)\frac{\varphi_{1}-\varphi_{2}}{2}=0 \end{array}\right..$$
where $b=\varepsilon_{1}\omega_{0}^{\mu -2}\sin\frac{\alpha \pi }{2},c=\varepsilon_{1}\omega_{0}^{\mu -2}\cos\frac{\alpha \pi }{2}$, and the following *b* and *c* express the same meaning.

To compare the slow variable model (12) with the *z_j_* obtained from the original system (5), the slow variable obtained from (12) is *φ_j_*, to be converted into a response in system (5). According to (7), *z_j_* can be determined by *φ_j_*, the formula is *z_j_* = *imag*(*φ_j_e^iτ^*), *z^′^_j_* = *real*(*φ_j_e^iτ^*), *j* = 1,2. The fractional derivative term in the original system (5) is solved by using the improved Oustaloup’s method [33].

Figure 2 (*ε*_1_ = 0) and Figure 3 (*ε*_1_ = 0.05) are displacement responses: (a) main structure (*z*_1_), (b) NES (*z*_2_), respectively (other parameters: *ε* = 0.05, *μ* = 1.9, *ω*_0_ = 2.5, *ξ_s_* = 0.01, *ξ_n_* = 0.005, *f*_e_ = 0.01, *κ* = 1). It can be seen from Figure 2 and Figure 3 that the responses of the slow varying model are in good agreement with the numerical solution in an abbreviated period at the beginning, and has a small deviation from the numerical solution after a long period of time. In general, the SVDE (12) can reflect the change trend of the original system response, and can be used for the comparison of the later stable responses.

### 3.2. The Discriminant Derivation of the Number of Equilibrium Points of S.1

To discuss the steady-state characteristics of the SVDE (S.1), it is difficult to calculate the equilibrium point of Equation (12) by directly making the derivative term 0. Therefore, the following variables are substituted for Equation (12):(13)$$\Psi_{1}=\varphi_{1},\Psi_{2}=\varphi_{1}-\varphi_{2}$$

Substituting Equation (13) into Equation (12), the following relation can be obtained:(14)$$\left\{ \begin{array}{l} {\dot{\Psi }}_{1}+\varepsilon \left({\dot{\Psi }}_{1}-{\dot{\Psi }}_{2}\right)+\xi_{s}\frac{\Psi_{1}}{2}+i\varepsilon \frac{\Psi_{1}-\Psi_{2}}{2}=\frac{f_{e}}{2}\\ \left(\varepsilon +\alpha b\right)\left({\dot{\Psi }}_{1}-{\dot{\Psi }}_{2}\right)-\alpha b{\dot{\Psi }}_{1}+i\varepsilon \frac{\Psi_{1}-\Psi_{2}}{2}-\left(c+bi+\xi_{n}-\frac{3\kappa }{4}i{\left|\Psi_{2}\right|}^{2}\right)\frac{\Psi_{2}}{2}=0 \end{array}\right..$$

If the derivative term in the above equation is zero, we obtain the equilibrium equation of the system as follows:(15a)$$\frac{9}{16}\kappa^{2}{\left|\Psi_{2}\right|}^{6}-\gamma_{3}\frac{3\kappa }{2}{\left|\Psi_{2}\right|}^{4}+\left(\gamma_{2}{}^{2}+\gamma_{3}{}^{2}\right){\left|\Psi_{2}\right|}^{2}=\gamma_{1}{}^{2},$$
(15b)$$\Psi_{2}=\frac{\gamma_{1}}{\left(\gamma_{2}+\gamma_{3}i-\frac{3\kappa }{4}i{\left|\Psi_{2}\right|}^{2}\right)},$$
and
(15c)$$\Psi_{1}=\frac{f_{e}}{\xi_{s}+i\varepsilon }+\frac{i\varepsilon }{\xi_{s}+i\varepsilon }\Psi_{2},$$
where
$$\gamma_{1}=\frac{\varepsilon }{\sqrt{\xi_{s}^{2}+\varepsilon^{2}}}f_{e},\gamma_{2}=\left(\frac{\xi_{s}\varepsilon^{2}}{\xi_{s}^{2}+\varepsilon^{2}}+c+\xi_{n}\right),\gamma_{3}=\left(\frac{\xi_{s}^{2}\varepsilon }{\xi_{s}^{2}+\varepsilon^{2}}+b\right).$$

Equations (15a)–(15c) are a set of uncoupled nonlinear equations, and the equilibrium points of the system can be calculated from top to bottom. It can be seen from Equations (15a)–(15c) that (15b) and (15c) have only a single real root, so the number of equilibrium points of the system is determined by the number of real roots of (15a). The number of real roots of the polynomial can be determined by the Cardano discriminant. For equation $a_{2}x^{3}+a_{1}x+a_{0}=0$, the Cardano discriminant is:(16)$$\Delta ={\left(\frac{a_{0}}{2a_{2}}\right)}^{2}+{\left(\frac{a_{1}}{3a_{2}}\right)}^{3}.$$

When Δ > 0, (15a) has a real root. When Δ = 0, (15a) has three real roots, of which at least two are equal. When Δ < 0, (15a) has three unequal real roots.

To determine the number of real roots of (15a), it is necessary to replace them with variables, as shown below:(17)$${\left|\Psi_{2}\right|}^{2}=\beta_{v}+\frac{\gamma_{3}\frac{3\kappa }{2}}{3\frac{9}{16}\kappa^{2}}=\beta_{v}+\frac{8\gamma_{3}}{9\kappa }.$$

Then, (15a) can be rewritten as:(18)$$\frac{9}{16}\kappa^{2}\beta_{v}^{3}+\left(\gamma_{2}^{2}-\frac{\gamma_{3}^{2}}{3}\right)\beta_{v}+\frac{8}{9\kappa }\gamma_{3}\left(\gamma_{2}^{2}+\frac{1}{9}\gamma_{3}^{2}\right)-\gamma_{1}^{2}=0.$$

Therefore:(19)$$\Delta ={\left(\frac{64\gamma_{3}\left(\gamma_{2}^{2}+\frac{1}{9}\gamma_{3}^{2}\right)-72\gamma_{1}{}^{2}\kappa }{81\kappa^{3}}\right)}^{2}+{\left(\frac{16\left(\gamma_{2}^{2}-\frac{\gamma_{3}^{2}}{3}\right)}{27\kappa^{2}}\right)}^{3}.$$

Particularly, when Δ = 0, the following relation can be obtained:(20)$$\kappa_{1,2}=\frac{8\left(\left(9\gamma_{2}^{2}+\gamma_{3}^{2}\right)\gamma_{3}\pm \sqrt{{\left(\gamma_{3}^{2}-3\gamma_{2}^{2}\right)}^{3}}\right)}{81\gamma_{1}^{2}}.$$

Obviously, the number of *κ* satisfying Δ = 0 is determined by the positive and negative of (21) as follows: (21)$$\Delta_{1}=\gamma_{3}^{2}-3\gamma_{2}^{2}$$

Δ_1_ is the judgment condition for the quantity whose Δ is equal to 0. When Δ_1_ = 0, Δ = 0, the system has three equilibrium points, and in other positions when Δ > 0, the system has only one equilibrium point. When Δ_1_ > 0, Δ has two zero points, and there are three equilibrium points in the system between the two zero points, and there is only one equilibrium point at other positions. When Δ_1_ < 0, Δ has no zero points, and there is only one equilibrium point at all positions.

### 3.3. The Discriminant Derivation of the Number of Equilibrium Points of S.2

The derivation method of S.2 is the same as S.1, so the derivation process is simplified, and the complex variables are substituted into (6) to obtain the SVDE of S.2 at 1:1 internal resonance, as shown below:(22)$$\left\{ \begin{array}{l} {\varphi^{\prime }}_{1}+\left(\varepsilon +\alpha b\right){\varphi^{\prime }}_{2}+\xi_{s}\frac{\varphi_{1}}{2}+\left(c+\left(\varepsilon +b\right)i\right)\frac{\varphi_{2}}{2}=\frac{f_{e}}{2}\\ \left(\varepsilon +\alpha b\right){\varphi^{\prime }}_{2}+\left(c+\left(\varepsilon +b\right)i\right)\frac{\varphi_{2}}{2}-\left(\xi_{n}-\frac{3\kappa }{4}i{\left|\varphi_{1}-\varphi_{2}\right|}^{2}\right)\frac{\varphi_{1}-\varphi_{2}}{2}=0 \end{array}\right.$$

Selecting the same variable as in the previous section to replace (Ψ_1_ = *φ*_1_, Ψ_2_ = *φ*_1_ − *φ*_2_), and the SVDE of S.2 is rewritten as:(23)$$\left\{ \begin{array}{l} {\dot{\Psi }}_{1}+\left(\varepsilon +\alpha b\right)\left({\dot{\Psi }}_{1}-{\dot{\Psi }}_{2}\right)+\xi_{s}\frac{\Psi_{1}}{2}+\left(c+\left(\varepsilon +b\right)i\right)\frac{\left(\Psi_{1}-\Psi_{2}\right)}{2}=\frac{f_{e}}{2}\\ \left(\varepsilon +\alpha b\right)\left({\dot{\Psi }}_{1}-{\dot{\Psi }}_{2}\right)+\left(c+\left(\varepsilon +b\right)i\right)\frac{\left(\Psi_{1}-\Psi_{2}\right)}{2}-\left(\xi_{n}-\frac{3\kappa }{4}i{\left|\Psi_{2}\right|}^{2}\right)\frac{\Psi_{2}}{2}=0 \end{array}\right.$$

If the derivative term in (23) is 0, the equilibrium point equations of S.2 can be obtained as follows:(24a)$$\frac{9}{16}\kappa^{2}{\left|\Psi_{2}\right|}^{6}-\frac{3}{2}{\bar{\gamma }}_{3}\kappa {\left|\Psi_{2}\right|}^{4}+{\bar{\gamma }}_{2}{\left|\Psi_{2}\right|}^{2}={\bar{\gamma }}_{1}$$
(24b)$$\Psi_{2}=\frac{K_{1}f_{e}}{\xi_{s}\left(\xi_{n}-\frac{3\kappa }{4}{\left|\Psi_{2}\right|}^{2}i\right)\left(c-B_{1}i\right)+\left(C_{1}-\frac{3\kappa }{4}{\left|\Psi_{2}\right|}^{2}i\right)K_{1}}$$
(24c)$$\Psi_{1}=\frac{\xi_{n}+c+\left(B_{1}-\frac{3\kappa }{4}{\left|\Psi_{2}\right|}^{2}\right)i}{c+B_{1}i}\Psi_{2}$$
where $B_{1}=\left(\varepsilon +b\right),C_{1}=\xi_{s}+\xi_{n},K_{1}=c^{2}+B_{1}^{2},$${\bar{\gamma }}_{1}=\frac{K_{1}^{2}}{\xi_{s}^{2}B_{1}^{2}+{\left(K_{1}+c\xi_{s}\right)}^{2}}f_{e}^{2}$,${\bar{\gamma }}_{2}=\frac{\xi_{s}^{2}\xi_{n}^{2}B_{1}^{2}+{\left(\xi_{s}\xi_{n}c+K_{1}C_{1}\right)}^{2}}{\xi_{s}^{2}B_{1}^{2}+{\left(K_{1}+c\xi_{s}\right)}^{2}}$, ${\bar{\gamma }}_{3}=\frac{\xi_{s}^{2}B_{1}K_{1}}{\xi_{s}^{2}B_{1}^{2}+{\left(K_{1}+c\xi_{s}\right)}^{2}}$.

To judge the number of real roots of (24a), comparing (24a) with (15a) to find that the same variable can be used to replace. Substituting (17) into (24a), (24a) can be rewritten as follows:(25)$$\frac{9}{16}\kappa^{2}\beta_{v}^{3}+\left({\bar{\gamma }}_{2}-\frac{4{\bar{\gamma }}_{3}^{2}}{3}\right)\beta_{v}+\frac{8{\bar{\gamma }}_{2}{\bar{\gamma }}_{3}}{9\kappa }-\frac{64{\bar{\gamma }}_{3}^{3}}{81\kappa }-{\bar{\gamma }}_{1}=0$$

The discriminant is obtained by substituting the coefficient of (25) into (16):(26)$$\Delta ={\left(\frac{8\left(72{\bar{\gamma }}_{2}{\bar{\gamma }}_{3}-81\kappa {\bar{\gamma }}_{1}-64{\bar{\gamma }}_{3}^{3}\right)}{729\kappa^{3}}\right)}^{2}+{\left(\frac{\left(48{\bar{\gamma }}_{2}-64{\bar{\gamma }}_{3}^{2}\right)}{81\kappa^{2}}\right)}^{3}$$

Similarly, assuming Δ = 0, the expression of *κ* can be simplified as follows:(27)$$\kappa_{1,2}=\frac{8\left(9\gamma_{2}-8\gamma_{3}^{2}\right)\gamma_{3}\pm 8\sqrt{{\left(4\gamma_{3}^{2}-3\gamma_{2}\right)}^{3}}}{81\gamma_{1}}$$

Therefore, the discriminant condition for the number of *κ* is:(28)$$\Delta_{2}=4{\bar{\gamma }}_{3}^{2}-3{\bar{\gamma }}_{2}$$

To understand the influence of various parameters on the number of equilibrium points when there is no inerter (i.e., *b_n_* = 0), according to Equations (2) and (3), the dynamic equations of the two schemes are the same. Therefore, taking S.2 as an example, this paper gives the discriminant Δ_2_ when *b_n_* = 0 by replacing parameters, as shown below:(29)$$\Delta_{2}=\frac{4\xi_{s}^{4}\varepsilon^{2}-3\left[{\left(\xi_{s}\xi_{n}\right)}^{2}+{\left(\xi_{n}+\xi_{s}\right)}^{2}\varepsilon^{2}\right]\left(\xi_{s}^{2}+\varepsilon^{2}\right)}{\varepsilon^{2}{\left(\xi_{s}^{2}+\varepsilon^{2}\right)}^{2}}$$

Observing the above formula, the value of Δ_2_ is determined by *ξ_n_*, *ξ_s_*, and *ε*, and the clear relationship is that the smaller *ξ_n_* is, the larger Δ_2_ is.

### 3.4. Stability Analysis Based on Lyapunov Theory

The equilibrium obtained from the equations derived in the previous section may be unstable. If the equilibrium point is unstable, the system may have a weakly modulated response (WMR) or SMR. As the response type of the system has a very important impact on the vibration suppression effect, it is necessary to analyze the stability of the equilibrium point of the system.

For S.1, the Equation (14) is expanded near the equilibrium point, i.e.,
(30)$$\Psi_{1}=\Psi_{10}+\delta_{1},\Psi_{2}=\Psi_{20}+\delta_{2}$$
where Ψ_10_ and Ψ_20_ are the equilibrium points of the system, and *δ*_1_ and *δ*_2_ are the small increments near the corresponding equilibrium points. After algebraic operation, the linear differential equations about the increment can be obtained as follows:(31)$$\left\{ \begin{array}{l} {\dot{\delta }}_{1}=-\xi_{s}\frac{\left(\varepsilon +\alpha b\right)}{ϒ}\frac{\delta_{1}}{2}-\frac{\alpha b}{ϒ}i\varepsilon \frac{\delta_{1}-\delta_{2}}{2}-\frac{\varepsilon }{ϒ}\left(c+bi+\xi_{n}\right)\frac{\delta_{2}}{2}+\frac{\varepsilon }{ϒ}\frac{3\kappa }{8}i\left(2{\left|\Psi_{20}\right|}^{2}\delta_{2}+\Psi_{20}^{2}\delta_{2}^{*}\right)\\ {\dot{\delta }}_{2}=-\xi_{s}\frac{\varepsilon }{ϒ}\frac{\delta_{1}}{2}+\frac{i\varepsilon }{ϒ}\frac{\delta_{1}-\delta_{2}}{2}-\frac{1+\varepsilon }{ϒ}\left(c+bi+\xi_{n}\right)\frac{\delta_{2}}{2}+\frac{1+\varepsilon }{ϒ}\frac{3\kappa }{8}i\left(2{\left|\Psi_{20}\right|}^{2}\delta_{2}+\Psi_{20}^{2}\delta_{2}^{*}\right) \end{array}\right.$$

By substituting the following formula into (31):(32)$$\Psi_{20}=N_{20}e^{i\theta_{20}},\delta_{1}=p_{1}+q_{1}i,\delta_{2}=p_{2}+q_{2}i,\delta_{2}^{*}=p_{2}-q_{2}i$$
and separating the real part from the imaginary part, the following relationship can be obtained:(33)$$\left\{ \begin{array}{l} {\dot{p}}_{1}=\frac{\varepsilon }{2ϒ}\left(-\xi_{s}\left(1+b\alpha /\varepsilon \right)p_{1}+b\alpha q_{1}-{ϒ}_{2}p_{2}+\left({ϒ}_{3}-b\alpha \right)q_{2}\right)\\ {\dot{q}}_{1}=\frac{\varepsilon }{2ϒ}\left(-b\alpha p_{1}-\xi_{s}\left(1+b\alpha /\varepsilon \right)q_{1}+\left(b\alpha -{ϒ}_{4}\right)p_{2}-{ϒ}_{1}q_{2}\right)\\ {\dot{p}}_{2}=\frac{\varepsilon }{2ϒ}\left(-\xi_{s}p_{1}-q_{1}-\left(1+1/\varepsilon \right){ϒ}_{2}p_{2}+\left(1+\left(1+1/\varepsilon \right){ϒ}_{3}\right)q_{2}\right)\\ {\dot{q}}_{2}=\frac{\varepsilon }{2ϒ}\left(p_{1}-\xi_{s}q_{1}-\left(1+\left(1+1/\varepsilon \right){ϒ}_{4}\right)p_{2}-\left(1+1/\varepsilon \right){ϒ}_{1}q_{2}\right) \end{array}\right.$$
where ${ϒ}_{1}=c+\xi_{n}-\frac{3N^{2}\kappa \sin2\theta_{20}}{4},{ϒ}_{2}=c+\xi_{n}+\frac{3N^{2}\kappa \sin2\theta_{20}}{4},$ ${ϒ}_{3}=b-\frac{3}{4}(2-\cos2\theta_{20})\kappa N_{20}^{2},$${ϒ}_{4}=b-\frac{3}{4}\left(2+\cos2\theta_{20}\right)N_{20}^{2}\kappa$. 

*θ*_20_ in the above equation can be obtained from the following equation:(34)$$\theta_{20}=-i\ln\frac{\Psi_{2}}{\left|\Psi_{2}\right|}$$

Obviously, Equation (33) is a linear constant system, the stability can be directly determined by Lyapunov theory, and the characteristic factor is defined:(35)Ξ=max(Re(λi))

*λ*_i_ are the eigenvalues of the coefficient matrix on the right side of (33). According to Lyapunov theory, if Ξ < 0, the equilibrium point of the system is asymptotically stable; if Ξ = 0, the equilibrium point is critically stable; if Ξ > 0, the equilibrium point is unstable. If a pair of eigenvalues cross the imaginary axis when they reach a certain point with a certain parameter, then this point is also called the Hopf bifurcation point of the system.

The characteristic matrix of the system is as follows:(36)$$A_{1}=\frac{\varepsilon }{2ϒ}\left(\begin{array}{cccc} -\xi_{s}\left(1+b_{1}/\varepsilon \right) & b_{1} & -{ϒ}_{2} & {ϒ}_{3}-b_{1}\\ -b_{1} & -\xi_{s}\left(1+b_{1}/\varepsilon \right) & b_{1}-{ϒ}_{4} & -{ϒ}_{1}\\ -\xi_{s} & -1 & -\left(1+1/\varepsilon \right){ϒ}_{1} & 1+\left(1+1/\varepsilon \right){ϒ}_{3}\\ 1 & -\xi_{s} & -1-\left(1+1/\varepsilon \right){ϒ}_{4} & -\left(1+1/\varepsilon \right){ϒ}_{2} \end{array}\right)$$

Therefore, according to the following formula: (37)$$\left|\lambda I_{4\times 4}-A_{1}\right|=0$$
its characteristic polynomial can be obtained as follows: (38)$$a_{1}\lambda^{4}+a_{2}\lambda^{3}+a_{3}\lambda^{2}+a_{4}\lambda^{1}+a_{5}=0$$

Since the forms of *a_i_* (*i* = 1, 2, …5) are too complex, and the eigenvalues in (38) can be easily calculated in MATLAB, the specific forms are not given here. For S.2, using the same method above, the system matrix *A*_2_ of its incremental state equation can be derived from (23) as follows:(39)$$A_{2}=\left(\begin{array}{cccc} -\frac{1}{2}\xi_{s} & 0 & -{ℜ}_{2} & {ℜ}_{3}\\ 0 & -\frac{\xi_{s}}{2} & {ℜ}_{4} & {ℜ}_{1}\\ \frac{c-ℜ\xi_{s}}{2ℜ} & -\frac{b+\varepsilon }{2ℜ} & -\frac{c}{2ℜ}-\frac{1+ℜ}{ℜ}{ℜ}_{2} & \frac{b+\varepsilon }{2ℜ}+\frac{1+ℜ}{ℜ}{ℜ}_{3}\\ \frac{b+\varepsilon }{2ℜ} & \frac{c-ℜ\xi_{s}}{2ℜ} & -\frac{b+\varepsilon }{2ℜ}+\frac{1+ℜ}{ℜ}{ℜ}_{4} & \frac{1}{-2ℜ}+\frac{1+ℜ}{ℜ}{ℜ}_{1} \end{array}\right)$$
where $ℜ=\left(b\alpha +\varepsilon \right),{ℜ}_{1}=\frac{3N^{2}\kappa \sin2\theta -4\xi_{n}}{8},{ℜ}_{2}=\frac{3N^{2}\kappa \sin2\theta +4\xi_{n}}{8},{ℜ}_{3}=\frac{3N^{2}\kappa \left(\cos2\theta -2\right)}{8},{ℜ}_{4}=\frac{3\kappa N^{2}\left(\cos2\theta +2\right)}{8}$.

The stability of the equilibrium point of S.2 is determined by the eigenvalue of the characteristic equation corresponding to the system matrix *A*_2_.

## 4. Numerical Simulation and Discussion

Since the cubic stiffness of the NES is not much different from the linear stiffness of the actual structure, this section only studies the characteristics of the system equilibrium point when *κ* changes within a certain range, but the method used is not limited by parameter changes. The system parameters discussed in this section are normalized parameters, in which the constant parameters *m_s_* = 10, *k_s_* = 0.6, and *f*_e_ = 0.01.

### 4.1. Influence of Parameters on the Number of Equilibrium Point

To find the parameter combination of three equilibrium points in the system without inerter (*b_n_* = 0), Figure 4a–c are drawn according to Δ_2_ (28), where Figure 4a shows that the larger *c_n_*, the smaller Δ_2_, which is consistent with the previous prediction. Figure 4b shows that when *m*_2_ increases, Δ_2_ first increases and then decreases, while the Figure 4c shows that the influence of *c_s_* is just opposite to *m*_2_. The greater the *c_s_*, Δ_2_ first decreases and then increases. To make Δ_2_ > 0, *m*_2_ should choose a small value, and *c_s_* should choose a relatively large value.

Then, selecting the appropriate parameters, and considering the NES of parallel and series inerter, the effects of parameters *μ* and *b_n_* on Δ_1_ and Δ_2,_ respectively, are discussed. Considering the influence of inerter (*b_n_* ≠ 0), for S.1, Figure 5a–c are drawn according to Δ_1_ (21), as shown below. 

Figure 5a shows that the two surfaces almost coincide, and *m*_2_ has an insignificant effect on Δ_1_, which is in line with the prediction of the curve when *c_s_* = 0.003 in Figure 4b. In Figure 5b, the change of *μ* directly changes the slope of Δ_1_ with *b_n_*, especially when *μ* = 5/3, the slope is 0. As shown in Figure 5c, *b_n_* has no effect on the *μ* range of Δ_1_ > 0, which is always between *μ* > 5/3 and 2. However, the larger *b_n_* is, the larger Δ_1_ is when *μ* > 5/3. In addition, when there is no inerter (*b_n_* = 0), the slope is 0, and Δ_1_ remains unchanged and less than 0, the system does not have multiple equilibrium points. Therefore, in the following discussion, *μ* should be selected between (5/3,2).

When S.2 adopt the same parameters as S.1, Δ_2_ is always negative, indicating that it is not easy to make Δ_2_ > 0 in S.2. To highlight the role of the inerter, for S.2, *m*_2_ = 0, and the influence of the inerter parameters on the value of Δ_2_ is analyzed. Under the premise of *m*_2_ = 0, the surfaces of Δ_2_ versus *μ* and *b_n_* is drawn for different *c_s_* (*c_s_* = 0.1, *c_s_* = 0.3) in Figure 6a; Figure 6b shows the curve of Δ_2_ versus *b_n_* for different *μ* (*c_s_* = 0.3); Figure 6c shows the curve of Δ_2_ versus *μ* for different *b_n_* (*m*_2_ = 0, *c_n_* = 0.002, *b_n_* = 0.4), and other parameters are the same as those in S.1.

As can be seen from Figure 6a,c, in the case of other parameter phase diagrams, the larger the *c_s_*, the larger the range of Δ_2_ greater than zero. It can be seen from Figure 6b that the change law of the curve is consistent with that of Figure 4b, which shows that the inertance (*b_n_*) of the series inerter can nominally replace *m*_2_. However, for S.1 the change law is different from that of the curve in Figure 4b, which shows that *b_n_* cannot simply replace *m*_2_ when the inerter is connected in parallel.

### 4.2. Verification of Equilibrium Point Discriminant

To verify the positive and negative correspondence between the number of equilibrium points and Δ, and the prediction results of Δ_1_ and Δ_2_ in the previous section, the curves of Δ and |Ψ_2_| with *κ* are drawn.

For S.1, the values of *m*_2_ = 0.3 and *m*_2_ = 4 are selected, and the other parameters are consistent with those in Figure 5b (*c_s_* = 0.003, *c*_n_ = 0.002, *b_n_* = 0.4). The curves of Δ (Equation (19)) and |Ψ_2_|of S.1 versus *μ* under different *κ* are drawn, as shown in Figure 7 and Figure 8.

In S.2, the parameters of Figure 6b (*m*_2_ = 0, *c_s_* = 0.3, *b_n_* = 0.4, *c_n_* = 0.002) and Figure 6c (*m*_2_ = 0, *c_s_* = 0.1, *b_n_* = 0.4, *c_n_* = 0.002) in the previous section are selected, and the curves of (a) Δ (Equation (26)) and (b) |Ψ_2_| of S.2 versus *κ* for different *μ* are drawn, as shown in Figure 9 and Figure 10.

Whether in S.1 or S.2, the positive and negative of Δ correspond to the number of equilibrium points of the system one by one, and can also correspond to the positive and negative prediction of Δ_1_ or Δ_2_ in the previous section. Under the same parameters, the smaller *μ*, the smaller the amplitude of the equilibrium point, and the less likely it is to switch the number of equilibrium points, indicating that the inertia and damping effects of the inerter exist simultaneously when *μ* is between 1 and 2. Here, *κ*_1,2_ = (0.22, 0.34) is calculated by substituting the parameters into (20) when *μ* = 1.8 in Figure 7b, and *κ*_1,2_ = (0.77, 1.26) is calculated by substituting the parameters into (27) when *μ* = 2 in Figure 9b. Obviously, the calculated results of the two groups of parameters are consistent with the positions of the actual simulation curves. In addition, as predicted in the previous section, it is not easy to produce multiple equations with S.2. Particularly, when Δ_1_ = 0 or Δ_2_ = 0, the system is always only one equilibrium point. It can be considered that the system is just in a critical state.

### 4.3. Stability Discrimination of Equilibrium Points

In this section, the Lyapunov method in Section 3.3 is used to judge the stability of the equilibrium points of S.1 and S.2, and to verify whether the calculation results of the equilibrium point are consistent with those calculated by the numerical algorithm. Of course, this is meaningful only when the equilibrium point is asymptotically stable.

Taking S.1 as an example, the verification method for slowly varying parameters is established. According to (7), after the equilibrium point of system (14) is calculated by (15), the state of the original system (5) corresponding to the equilibrium point when *t* = 0 can be expressed as:(40)[z1,z′1,z2,z′2]=[Im(φ1),Re(φ1),Im(φ2),Re(φ2)]

A small disturbance (*δ*_1_, *δ*_2_, *δ*_3_, *δ*_4_) is added to (40) and brought into the original system (5) as an initial condition. If the equilibrium point of the system is stable, the time response is stable after a period.

If Ψ_2_ is expanded in the complex plane with *N*_2_*e**^iθ^*, then |Ψ_2_| = *N*_2_. The steady-state amplitude |Ψ_2_| is obtained from (15a). Since Ψ_2_ = *φ*_1_ − *φ*_2_, the response of the original system can be converted to Ψ_2_, that is, Ψ_2_ = −*z′*_1_ − *z′*_2_ + (*z*_1_ − *z*_2_)*i*. The original system (5) is calculated by the fourth order Runge–Kutta (R–K) method, and its steady-state value *N*_2_ can be obtained through conversion, as shown below:(41)$$N_{2}={\langle \sqrt{{\left(z_{1}-z_{2}\right)}^{2}+{\left({\dot{z}}_{1}-{\dot{z}}_{2}\right)}^{2}}\rangle }_{\left(\tau_{1},\tau_{2}\right)}$$
where < >_(*τ*1_, *_τ_*_2)_ represents the weighted average in the (*τ*_1_, *τ*_2_) period. Obviously, S.2 can also be treated in the same way.

Adding a disturbance *δ*_1_ = 0.01 to *z*_1_(0), the stability of the equilibrium point of the original system is analyzed. For S.1, the different stabilities of the equilibrium points are drawn, as shown in Figure 11, and the parameters of Figure 7b and Figure 8b are used in Figure 11a,b, respectively. For S.2, the different stabilities of the equilibrium points are drawn, as shown in Figure 12, and the parameters of Figure 9b and Figure 10b are used in Figure 12a,b, respectively. In Figure 11 and Figure 12, the red squares represents the stable equilibrium points obtained by the R–K method and the black dots represents the stable equilibrium points obtained by (15a). The other represent unstable equilibrium points are also represented.

Firstly, from the above four pictures, the stable equilibrium points obtained by the R–K method are the same as that obtained by the calculation in (15a) or (24a), which verifies the accuracy of the analytical method. No matter which scheme, when the system has three equilibrium points, the switching of the stability always occurs near the turning point of the equilibrium point curve, and Δ_1_ and Δ_2_ proposed in this paper can only predict the turning point position, and the parameters are few and easy to analyze.

Figure 11a,b show that both schemes have three equilibrium points, but Figure 11a shows two unstable equilibrium points with large amplitude and one stable equilibrium point with small amplitude, and Figure 11b shows an unstable equilibrium point sandwiched between the two stable equilibrium points, indicating that there are two different bifurcation phenomena in the system. When *b_n_* is calculated according to Δ_1_ = 0, or Δ_2_ = 0 is taken as the inertance of the system, the equilibrium point of the system is always stable with the change of κ. It is more difficult to produce multiple equilibrium points in S.2 (Figure 12) than S.1 (Figure 11), and the range of unstable parameters is relatively small. 

### 4.4. The Time-Domain Response 

Next, the time-domain responses are drawn to verify the accuracy of the above equilibrium stability analysis results, and the influence of equilibrium stability on the response type when the slow varying system has multiple solutions is explored. Particularly, to maintain the consistency between the steady-state amplitude *N*_2_ of the system and |Ψ_2_| in the previous section, “*z*_1_ − *z*_2_” is drawn instead of “*z*_2_”.

For S.1, we selected the parameters when k = 0.11 in Figure 11a (*μ* = 2 and *μ* = 1.9), and plotted the time-domain response of the original system (Figure 13 (*μ* = 2) and Figure 14 (*μ* = 1.9)) under different initial interference (*δ*_1_).

In Figure 13a–d, the corresponding initial amplitude|Ψ_2_|(0) is (0.11, 0.41, 0.56, 0.91), just between the three equilibrium points (|Ψ_2_|_1,2,3_ = 0.3109, 0.49, 0.79). The results show that the response can be stabilized at the small stable equilibrium point only when it is between the small equilibrium point and the intermediate equilibrium point, while the system response is in the SMR when other disturbances occur, which is consistent with the conclusion when *μ* = 2, *κ* = 0.11 in Figure 11a. The four *δ*_1_ are shown in Figure 14a–d, and the corresponding initial amplitudes (|Ψ_2_|(0)) are (0.07, 0.43, 0.73, 0.93) are just between the three equilibrium points (|Ψ_2_|_1,2,3_ = 0.24, 0.63, 0.81). If the initial amplitude is less than the second largest equilibrium point, the system response asymptotically tends to the small stable equilibrium point after a long time of WMR (as shown in Figure 14a,b), while when it is greater than the second largest equilibrium point, the system response is in the SMR (as shown in Figure 14c,d).

Then, the parameters (*m*_2_ = 4, *κ* = 0.1, *μ* = 2) and (*m*_2_ = 4, *κ* = 0.1, *μ* = 1.9) in Figure 11b are selected to draw the time-domain response of the original system (5) (as shown in Figure 15 and Figure 16). Figure 15a–d show that no matter what kind of initial interference, the system response tends to be stable after a long time of WMR. When the initial value is between two stable equilibrium points, it is always stable at the small equilibrium point, otherwise, it is stable at the large stable equilibrium point. The four *δ*_1_ are shown in Figure 16a–d, and the corresponding initial amplitudes (|Ψ_2_|(0)) are (0.06, 0.33, 0.73, 0.93) are just between the three equilibrium points (|Ψ_2_|_1,2,3_ = 0.24, 0.67, 0.84). Basically consistent with the situation of *μ* = 2 (Figure 15), the system response can be stabilized at two stable equilibrium points, but the time to reach the stable equilibrium point is much shorter, and there is no obvious modulated process.

For S.2: from Figure 10a,b in Section 4.2 and Figure 12a,b in Section 4.3, we can see that it is not easy to maintain multiple solutions, and the stability of the three equilibrium points is the same as that of S.1 (Figure 11b). To verify the special case of S.2, that is, whether the inerter can replace the mass function of traditional NES, we selected three groups of parameter schemes for comparison: T_1_ (*μ* = 2, *m*_2_ = 0, *b_n_* = 0.4), T_2_ (*b_n_* = 0, *m*_2_ = 0.4), T_3_ (*μ* = 1.9, *m*_2_ = 0, *b_n_* = 0.4) and the other parameters are selected when *κ* = 1.1 in Figure 12a.

Figure 17a,b, and c show the phase diagrams of the three parameter schemes of the original system (6). The initial interference selection method is the same as that of S.1, and the four initial interferences (*δ*_1_) are (−0.01, 0.0, 8, 0.14, 0.3). Figure 18a,b show the displacement responses (*z*_1_, *z*_1_ − *z*_2_) of T_1_ and T_2_ when the interference is −0.01 and 0.14, respectively. Figure 18c,d show the displacement (*z*_1_, *z*_1_ − *z*_2_) response of T_3_ and T_2_ when the interference is −0.01 and 0.14, respectively.

It can be seen from Figure 18c,d that the time-domain responses of T_3_ and T_2_ are basically coincidental, which shows that the *b_n_* of S.2 can replace the role of *m*_2_ equivalently when the integer order inerter is connected in series. From Figure 17a and Figure 18a,b, T_1_ and T_2_ can be stabilized at small stable equilibrium points no matter how large the interference (*δ*_1_) is. As can be seen from Figure 18b, when the initial amplitude is above the small equilibrium point, the amplitude of NES jumps from the large equilibrium point to the small equilibrium point due to the fractional order inerter in series in S.2, resulting in the inability to effectively suppress the displacement of the main structure. At this time, the effect of vibration suppression is not as good as that of series integer order inerter. 

The analysis of different situations of the above two schemes shows that the stability discrimination in the previous section is accurate. No matter the scheme, *μ* has a great impact on response, it is sensitive to the initial value when only the inertia effect (i.e., *μ* = 2) exists, and it is easy for various modulated responses to appear in the time-domain response. Under the same conditions, S.1 is more sensitive to the nonlinear parameter *κ* than S.2, and it is easy to produce different system responses.

## 5. Conclusions

In this paper, the inerter described by fractional derivative is used in traditional NES for the first time. Combined with the CA-X method and fractional properties, especially fractional Leibniz theorem, the difficulty of deriving SVDE from fractional nonlinear dynamic equations is solved. Through the verifications of analytical and numerical methods, the accuracy of SVDE ensures that the analysis is reliable.

The discriminants Δ_1_ (S.1) and Δ_2_ (S.2) for the number of equilibrium points are proposed. Compared with Δ, their expressions are simpler, which can directly and accurately predict the number of equilibrium points of the equilibrium point equation, and the range of *κ* corresponding to the system with three equilibrium points can be calculated by (20) and (27).

The verifications in Figure 11, Figure 12, Figure 13, Figure 14, Figure 15, Figure 16, Figure 17 and Figure 18 show that the Lyapunov theory given in this paper is accurate in judging the stability of the equilibrium point. For S.2, when *m*_2_ = 0, the introduction of the inerter does not increase the complexity of the structure, and the inerter can not only provide inertia, but also provide damping. S.1 has two unstable equilibrium points or two stable equilibrium points at the same time, so S.1 is more prone to various responses, especially various WMR and SMR. Compared with integer order inerter, the introduction of fractional order inerter, whether in series or in parallel, means that the amplitude of the equilibrium point on the NES vibrator is smaller, but it is also for this reason that it is not easy to produced modulated responses in S.2, and the vibration suppression effect of the main structure is not good. 

## Figures and Tables

**Figure 1 sensors-22-06408-f001:**
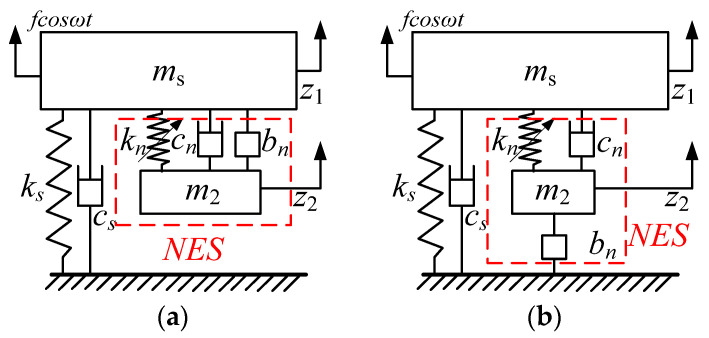
Two vibration models of the SDOF system coupled with NES: (**a**) (S.1); (**b**) (S.2).

**Figure 2 sensors-22-06408-f002:**
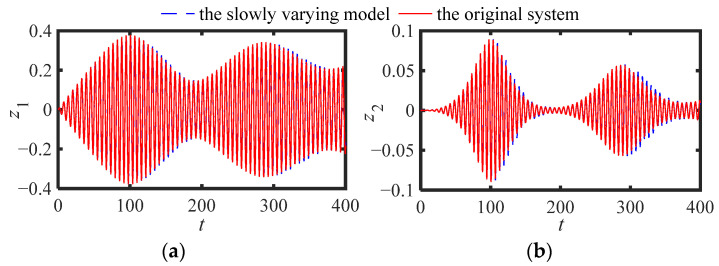
Displacement responses: (**a**) main structure, (**b**) NES (*ε*_1_ = 0).

**Figure 3 sensors-22-06408-f003:**
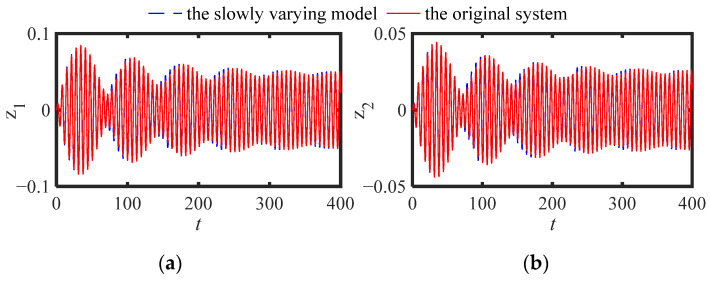
Displacement responses: (**a**) main structure, (**b**) NES (*ε*_1_ = 0.05).

**Figure 4 sensors-22-06408-f004:**
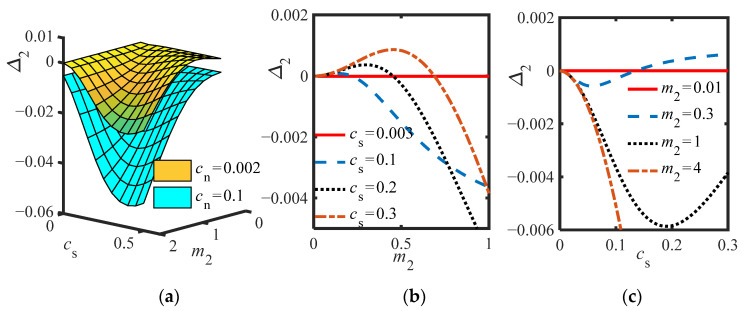
(**a**) Δ_2_ vs. *c_s_* and *m*_2_ at different *c_n_*; (**b**) Δ_2_ vs. *m*_2_ at different *c_s_* (*b_n_* = 0, *c_n_* = 0.002); (**c**) Δ_2_ vs. *c_s_* at different *m*_2_ (*b_n_* = 0, *c*_n_ = 0.002).

**Figure 5 sensors-22-06408-f005:**
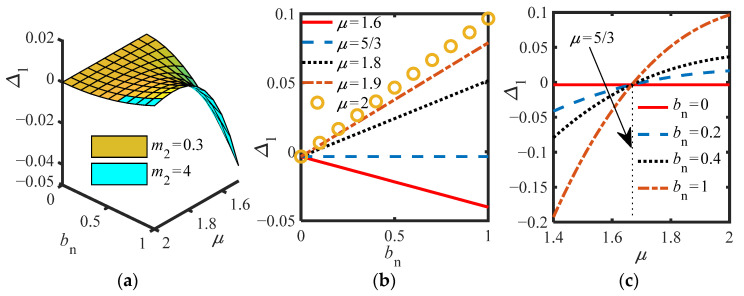
(**a**) Δ_1_ vs. *μ* and *b_n_* for different *m*_2_; (**b**) Δ_1_ vs. *b_n_* for different *μ* (*m*_2_ = 0.3, *c_s_* = 0.003, *c_n_* = 0.002); (**c**) Δ_1_ vs. *μ* for different *b_n_* (*m*_2_ = 0.3, *c_s_* = 0.003, *c_n_* = 0.002).

**Figure 6 sensors-22-06408-f006:**
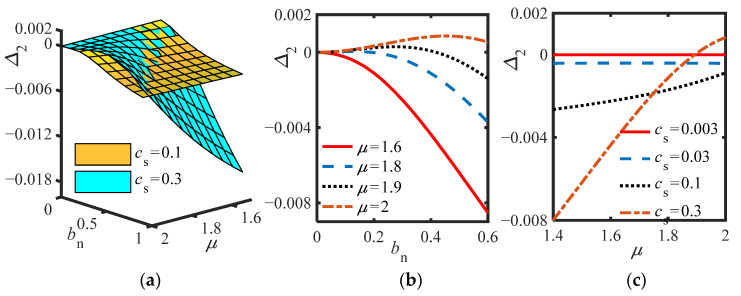
(**a**) Δ_2_ vs. *μ* and *b_n_* for different *c_s_*; (**b**) Δ_2_ vs. *b_n_* for different *μ* (*c_s_* = 0.3); (**c**) Δ_2_ vs. *b_n_* for different *c_s_* (*m*_2_ = 0, *c_n_* = 0.002, *b_n_* = 0.4).

**Figure 7 sensors-22-06408-f007:**
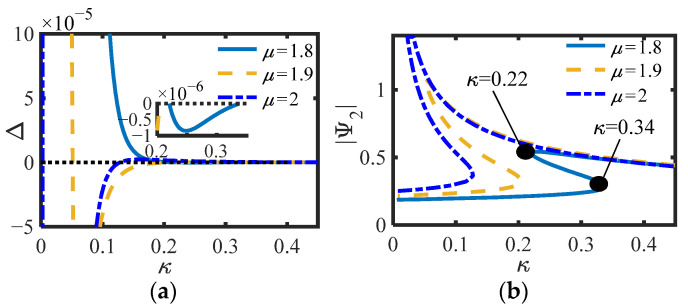
(**a**) Δ and (**b**)|Ψ_2_| of S.1 vs. *κ* for different *μ* (*m*_2_ = 0.3).

**Figure 8 sensors-22-06408-f008:**
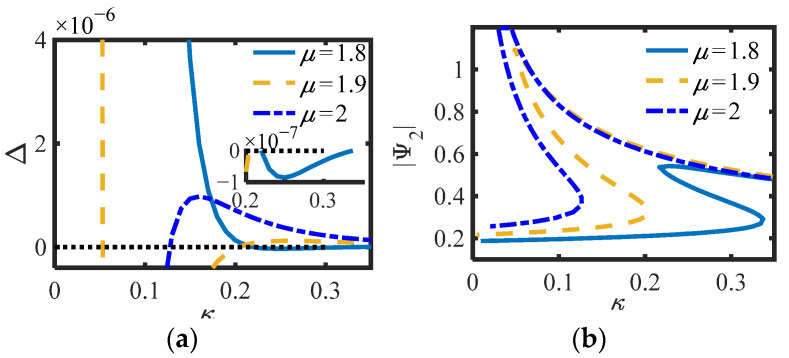
(**a**) Δ and (**b**)|Ψ_2_| of S.1 vs. *κ* for different *μ* (*m*_2_ = 4).

**Figure 9 sensors-22-06408-f009:**
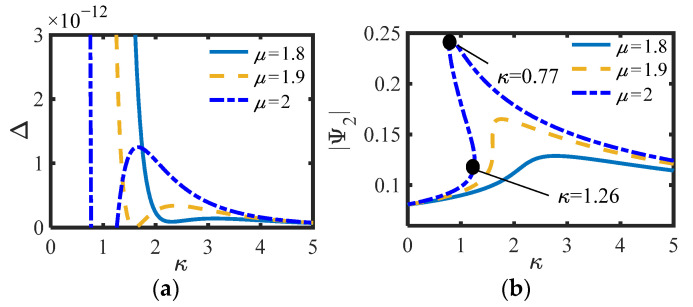
(**a**) Δ and (**b**) |Ψ_2_| of S.2 vs. *κ* for different *μ* (*c_s_* = 0.3).

**Figure 10 sensors-22-06408-f010:**
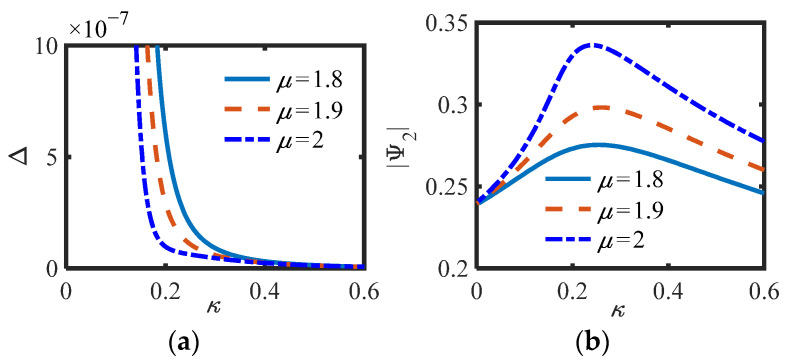
(**a**) Δ and (**b**) |Ψ_2_| of S.2 vs. *κ* for different *μ* (*c_s_* = 0.1).

**Figure 11 sensors-22-06408-f011:**
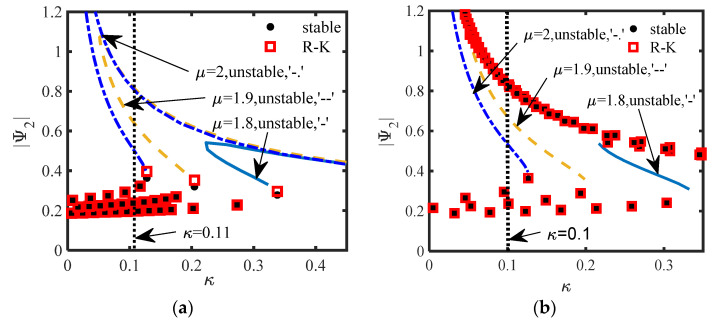
The different stabilities of the equilibrium points (S.1): (**a**) *m*_2_ = 0.3; (**b**) *m*_2_ = 4.

**Figure 12 sensors-22-06408-f012:**
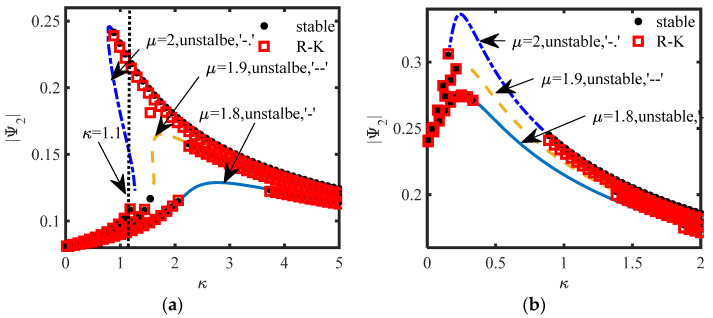
The different stabilities of the equilibrium points (S.2) (**a**) *c_s_* = 0.3; (**b**) *c_s_* = 0.1.

**Figure 13 sensors-22-06408-f013:**
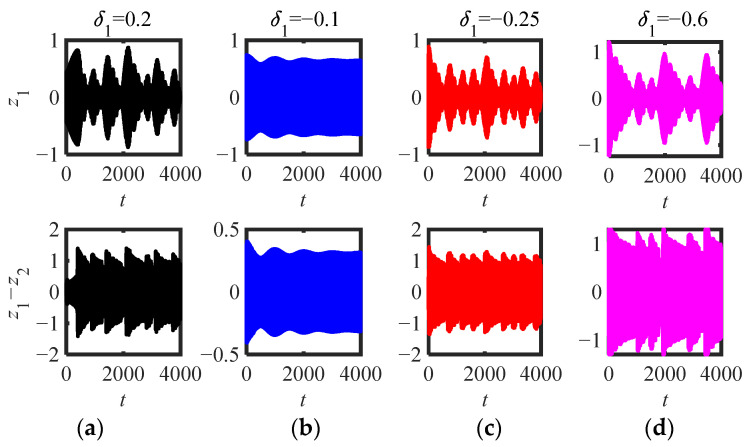
Time-domain response of (5) (parameters in Figure 11a *m*_2_ = 0.3, *μ* = 2, *κ* = 0.11): (**a**) *δ*_1_ = 0.2, (**b**) *δ*_1_ = −0.1, (**c**) *δ*_1_ = *−*0.25, (**d**) *δ*_1_ = −0.6.

**Figure 14 sensors-22-06408-f014:**
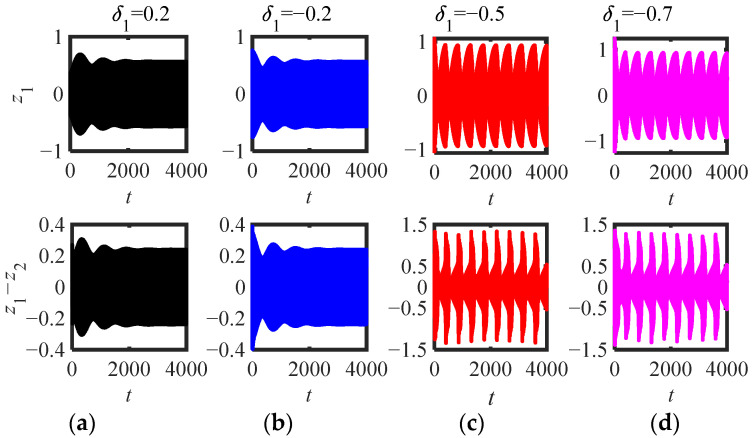
Time-domain response of (5) (parameters in Figure 11a *m*_2_ = 0.3, *μ* = 1.9, *κ* = 0.11): (**a**) *δ*_1_ = 0.2, (**b**) *δ*_1_ = −0.2, (**c**) *δ*_1_ = −0.5, (**d**) *δ*_1_ = −0.7.

**Figure 15 sensors-22-06408-f015:**
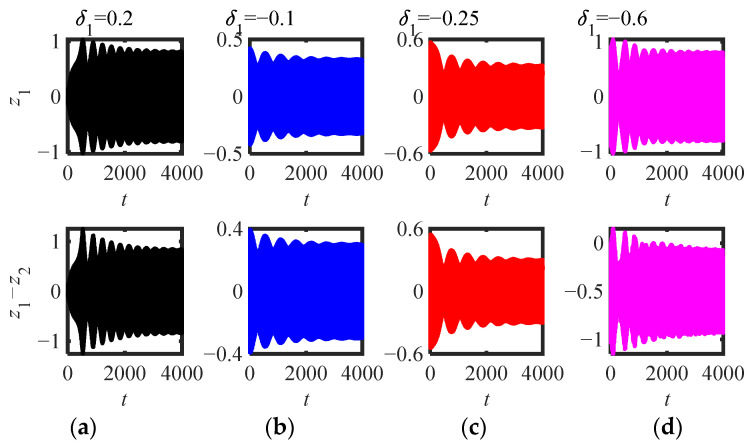
Time-domain response of (5) (parameters in Figure 11b *m*_2_ = 4, *μ* = 2, *κ* = 0.1): (**a**) *δ*_1_ = 0.2, (**b**) *δ*_1_ = −0.1, (**c**) *δ*_1_ = *−*0.25, (**d**) *δ*_1_ = −0.6.

**Figure 16 sensors-22-06408-f016:**
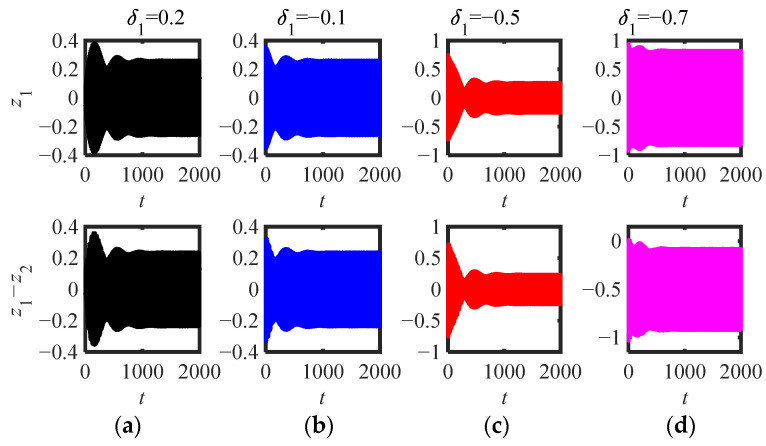
Time-domain response of (5) (parameters in Figure 11b *m*_2_ = 4, *μ* = 1.9, *κ* = 0.1): (**a**) *δ*_1_ = 0.2, (**b**) *δ*_1_ = −0.1, (**c**) *δ*_1_ = *−*0.5, (**d**) *δ*_1_ = −0.7.

**Figure 17 sensors-22-06408-f017:**
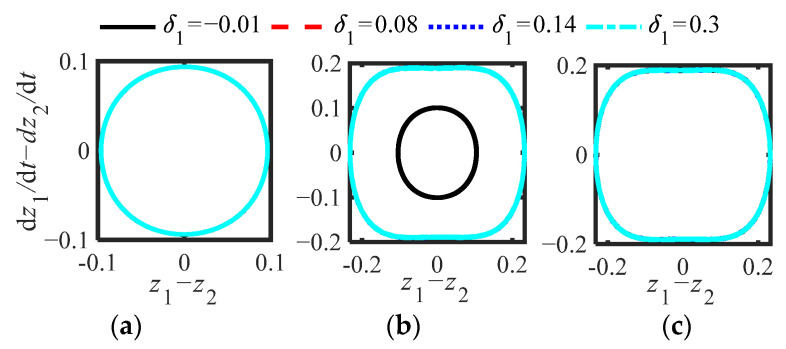
Phase diagrams (*κ* = 1.1) of (6) at different *δ*_1_: (**a**) T_1_, (**b**) T_2_, (**c**) T_3_.

**Figure 18 sensors-22-06408-f018:**
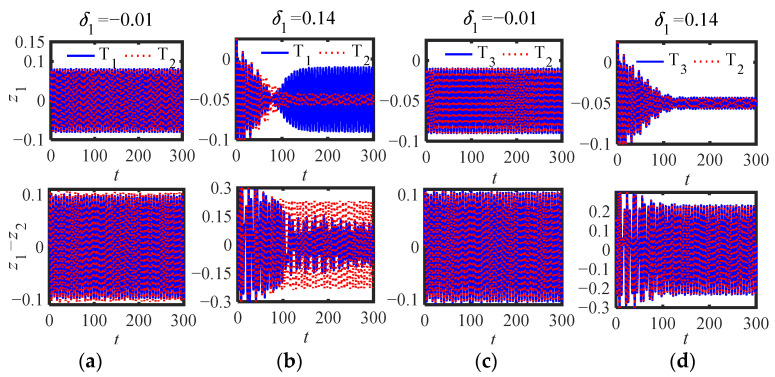
Time-domain responses (*κ* = 1.1) of (6) with different parameter schemes (**a**) T_1_ and T_2_ (*δ*_1_ = −0.01), (**b**) T_1_ and T_2_ (*δ*_1_ = −0.14), (**c**) T_3_ and T_2_ (*δ*_1_ = −0.01), (**d**) T_3_ and T_2_ (*δ*_1_ = −0.14).

## Data Availability

Not applicable.

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
