# Peer review of "Vibration Analysis of a 1-DOF System Coupled with a Nonlinear Energy Sink with a Fractional Order Inerter"

_sensors, 2022, doi:10.3390/s22176408_

Round 1
Reviewer 1 Report
In this paper, the effect of the NES with the fluid inerter described by fractional derivative model on the single degree of freedom system was studied in the manuscript. The equations of motion were solved analytically and numerically using the complex-averaging and Runge Kutta methods. The number of equilibrium points was determined using Cardano discriminant. Surveying following comments is recommended and may improve your manuscript:
1) According to equations of manuscript, it seems that you should change m1 to ms in figure 1.
2) Explain more about figure 16 (c) in the manuscript.
3) Comparing between the system responses without NES, with traditional NES, and with FNES in a single figure could be interesting and improve the manuscript level.
Reviewer 2 Report
This paper introduces the fluid inerter, described by fractional derivative model, into the traditional NES. Then, the dynamic equation is established and analyzed by the complex-averaging method, fractional Leibniz theorem, and the variable substitution method. The slowly varying characteristics, the stability and the inertial effect, etc, are discussed. The investigation is interesting. But before accepted for publication, improvememts are required.
1. In Abstract, SVDE, schemes 1 and 2 lack definition.
2. In Abstract and Introduction, it is claimed that this paper uses a fluid inerter to design a new NES, named FNES. But in the following modeling and analysis sections, the fluid inerter seems not being addressed. And the analyzed coupled system models are derived by using an ideal fractional-order inerter model, which is not significantly related to fluid inerter.
3. The concept of Target Energy Transfer is mainly applied to explain the mechanism that NES accelerates energy dissipation of the main system. That is for free vibration. For force vibration, the concept of Strongly Modulated Response is more suitable for analyzing the NES
4. Is αDαt in Eq. (10) correct? And the description of the Leibniz theorem of fractional calculus is unclear and makes it difficult for the reader to understand
5. In Figures 2 and 3, the red line has completely covered the blue line, which makes it impossible to distinguish the difference between the two curves.
6. Whether the discriminative law of ∆2 of the number of periodic solutions is the same as ∆1. That is seemingly not emphasized in the original text.
7. For forced vibrations, the appearance of the Strongly Modulated Response (SMR) means that the NES will present a better damping effect. And the appearance of SMR is related to the Neimark-Sacker bifurcation. Therefore, the bifurcation analysis is essential to understanding the damping mechanism of NES. However, in Section 4.3, the authors only analyzed the stability of the periodic solution and did not consider the bifurcation of the periodic solution.
8. There are some problems with the layout of the images (Figures 7-10) and the language grammar.
Reviewer 3 Report
Dear Authors,
This article has been submitted to Sensors journal, however, there are no references in the content. Therefore, you should change the journal or highlight the issues related to sensors.
Commonly used abbreviations cannot be changed, so the abbreviation NAS should remain. Abbreviations are not used before they are introduced.
Author Response
Point 1: This article has been submitted to Sensors journal, however, there are no references in the content. Therefore, you should change the journal or highlight the issues related to sensors.
Response 1: Thank you for your suggestions. We focused on the discussion of research methods before, and now a lot of improvements have been made in the revised draft. The overall idea is to discuss the advantages of the method proposed in this paper from the relationship between the output response and the stability of the equilibrium point. Firstly, the background introduction of the corresponding modulation response is added in the introduction. Secondly, the influence of stability on the response type is discussed in detail in Section 4 time domain simulation. Finally, the corresponding improvements are made in the conclusion and summary. By describing the influence of stability on the type of modulation response, it helps engineers understand how to design structural parameters and which type of vibration response is easy to get under harmonic excitation. Due to the influence of nonlinear parameters, the response is very sensitive to the initial value, so the research in this paper can also help engineers avoid the unfavorable initial value region in practical work.
Point 2: Commonly used abbreviations cannot be changed, so the abbreviation NAS should remain. Abbreviations are not used before they are introduced.
Response 2: Thank you for your valuable suggestions. We have improved the usage of abbreviations in the full text. For example, “SDOF” is changed to “1-DOF”, “FNES” is changed to “fractional-order NES”, “strong modulation response (SMR)” is changed to “strongly modulated response (SMR)”,and so on.
Additionally, we have deleted some repetitive presentation, and some grammar and spelling errors s in our revised paper have also been corrected.
1) The third conclusion is improved in the abstract, highlighting the discussion of stability to understand its impact on response types. In addition, the series integral order inerter can completely replace the role of mass. At the same time, it is pointed out that the performance of the series fractional order inerter is not necessarily better.
2) In Section 1, the background description is added, especially the description of fractional order model and the introduction of SMR related research.
3) In the conclusion of section 5, some conclusions are improved to highlight the discussion of stability to understand its impact on response types. In addition, the series integral order inerter can completely replace the role of mass. At the same time, it is pointed out that the performance of the series fractional order inerter is not necessarily better.
And supplementary notes on the modification of figures in our revised paper are as follows:
1)We adjusted the size of the graphs and the thickness of the lines in Figures 2 and 3.
2)We have redrawn Figures 7-10 and corrected the errors according to the journal template.
3) Figures 15-18 are improved, in which the initial interference that makes the initial equilibrium point smaller than the minimum equilibrium point replaces the zero initial to obtain the time domain response. In addition, the explanatory notes of these figures are written in more detail and with emphasis.
Round 2
Reviewer 2 Report
The manuscript is improved. I have no suggestion for further improvement.
Author Response
Thank you!
Reviewer 3 Report
Dear Authors,
you have not answered my most important question:
"This article has been submitted to Sensors journal, however, there are no references in the content. Therefore, you should change the journal or highlight the issues related to sensors."
To put it another way, where does the article mention the sensor and any effect of the sensor on the system under consideration?
Author Response
"Sensors are widely used in measurement, detection, automatic control and other aspects. However, the nonlinear relationship between the output of the sensor and the measured value has always troubled the application of sensors.
1) The vibration system studied in this paper generally needs to use the linear variable differential transformer (LVDT) displacement sensor to collect the displacement signal. At present, the displacement of most LVDT sensors and the measured value are nonlinear, which leads to the nonlinear error of the measured data. This error greatly reduces the accuracy of LVDT sensor measurement results. If the measured object has strong nonlinearity, it will further reduce the reliability of the measurement results. The analysis of the nonlinear characteristics of the measured system helps to reduce the complexity of the sensor nonlinear correction method.
2) The vibration system studied in this paper is an asymmetric nonlinear system with cubic nonlinear stiffness, and there is also a fluid inertial container with multiphase mechanical characteristics. It is challenging to understand the nonlinear dynamic characteristics of the system. It is difficult to study directly from the experiment, so it is important to analyze the influence of nonlinear parameters and newly introduced inertial vessel parameters on its response type and mechanism analysis from the theoretical point of view.
3) The results of this paper show that the response type is related to the system parameters, especially the nonlinear stiffness(κ) and the inerter parameters(μ,bn). Secondly, like other nonlinear systems, it is sensitive to the initial value. The influence of parameters on the two schemes is different, but the influence of nonlinear stiffness is greater. If the parameters of inerter are properly selected, and κ is within a certain range, the system will have strongly or weakly modulated response, and there is also a response like the linear system that is always stable. Through the study of the stability of the equilibrium point, it is found that the response type of the system is also related to the initial value near the different equilibrium points. There may be asymptotically stable responses near the stable equilibrium points, and strongly or weakly modulated response near the unstable equilibrium points. With this understanding of the system, it provides a theoretical basis for judging whether the data measured by the sensor is effective, so as to guide the engineers to select a reasonable sensor nonlinear correction method to improve the accuracy. "